# NEURAL VIDEO ENCODING

## ABSTRACT

Deep neural networks have had unprecedented success in computer vision, natural language processing, and speech largely due to the ability to search for suitable task algorithms via differentiable programming. In this paper, we borrow ideas from Kolmogorov complexity theory and normalizing flows to explore the possibilities of finding arbitrary algorithms that represent data. In particular, algorithms which encode sequences of video image frames. Ultimately, we demonstrate neural video encoded using convolutional neural networks to transform autoregressive noise processes and show that this method has surprising cryptographic analogs for information security.

In algorithmic information theory, the *Kolmogorov complexity* of an object is the length of the shortest computer program that, without additional input, produces the object as output (Ming & Paul, 2008). For example, a long sequence of ones contains very little information because a short program

   **for** $i \leftarrow 0, n$ **do** print 1

can output the data. Similarly, the transcendental number $\pi = 3.1415\ldots$ is an infinite sequence of seemingly random decimal digits but contains only a small amount of information since there exists a very short program that produces the consecutive digits of $\pi$ forever (Bailey et al., 1997). While this idea of exchanging data with computation is theoretically intriguing, from a practical point of view, explicit specification of instructions to obtain some target representation is generally an intractable approach for arbitrary data.

Recently, however, artificial intelligence had broad success in computer vision, natural language processing, and speech largely due to the ability to search for suitable task algorithms using differentiable programming together with large datasets to obtain abstract computations in the form of neural networks. In this paper, we turn the process around and leverage this same differentiable programming approach to find abstract computations which represent data. That is, we seek algorithms which encode arbitrary information for storage and retrieval. In particular, we demonstrate methods for the encoding of sequential video data within the weights of convolutional neural networks.

In the simplest formulation, video data, $X$, is treated as an finite ordered sequence of image frames

$$X = \{x_0, x_1, \ldots, x_N\} \tag{1}$$

and we seek an autoregressive computation of the form

$$y_{i+1} = f_\theta(y_i) \tag{2}$$

where $y_0 \leftarrow x_0$ and $f$ is some convolutional neural network parametrized by weights $\theta$ such that $y_i \approx x_i$ . It is possible that $f$ can be trained using a self-supervised learning approach where consecutive image frames form training examples and the sum-of-squares, $\|y_i - x_i\|^2$, is minimized. That is, during training, $f$ learns a mapping from $x_i$ to $x_{i+1}$, or rather, learns to advance the current frame by one step. The idea is somewhat similar to conventional autoencoders (Hinton & Zemel, 1993), but there, $f$ is trained with the identity mapping, from $x_i$ on to $x_i$. The catch here is that after training, during recall, $f$ does not receive $x_i$ as input but instead $y_i$ from the previous evaluation. Since $y_i$ only approximates $x_i$, reconstruction error accumulates at each iteration of eq. 2 until sequence reproduction fails[1].

The natural inclination at this point is to extend the approach to train over the full sequence by evaluating eq. 2 $N$ times to produce $\{y_1, \ldots, y_N\}$ and minimizing the mean squared error over the

---

[1]This problem is associated with *teacher forcing*. See, e.g., Goodfellow et al. (2016).

entire sequence at each training iteration. This way, how the network is trained and how the sequence is recalled post-training are congruent. Unfortunately, for sequences beyond a few elements, this is not tractable as long unrolled gradients readily vanish or explode and there is no opportunity for batching which significantly limits training performance and scalability.

The situation can be salvaged by adopting a *curriculum learning* strategy (Bengio et al., 2009) whereby the training sequence length is slowly increased after achieving some nominal loss objective over shorter intervals. It is important to note that, because CNNs have no recurrent hidden state, this approach can be successfully batched during training over random subsequences of $X$. Therefore, modulating the training subsequence length to one yields the aforementioned self-supervised learning approach whereas a training sequence length equal to the number of frame elements yields the full sequence loss. While this batched curriculum learning strategy does mitigate challenges associated with teacher forcing, gradients can still be temperamental for larger training sequence lengths necessitating the use of gradient clipping and learning rate scaling (You et al., 2017; 2019).

An interesting situation occurs when $x_i = x_j$ but $f(x_i) \neq f(x_j)$. That is, there exists a non-unique mapping or *sequence ambiguity* which is not resolvable from a purely supervised learning point of view since a unique input evaluates to multiple output. It is perhaps surprising to some readers that eq. 2 above is capable of handling this situation successfully, despite CNNs having no recurrent hidden state. Due to the lack of teacher forcing and high dimensionality of the data, the network learns to encode (and decode) state information within the pixels thus willfully incurring a local reconstruction penalty in order to facilitate sequence completion. The success of the batched sequence learning strategy depends on the nature of the ambiguity.

Just as the state of a dynamical system is not well defined by a single observation, often encoding is facilitated by sequence *dilation* of the learned mapping which more uniquely defines the trajectory of sequence evolution. A single frame mapping from $x_i$ to $x_{i+1}$ exhibits a dilation of $d = 1$ and can be naturally extended to $d = 2$ by channel concatenation to learn a computation of the form $[y_i; y_{i+1}] = f([y_{i-1}; y_i])$ with dimension $(2C) \times H \times W$; and so on.[2] This helps to individuate samples and thereby allay the need for contextual sequence information to achieve recall. For example, the alternating sequence ambiguity in $\{A, B, A, C, A, D\}$ is easily resolved by dilation $d = 2$ since all training examples, $f(AB) \to BA$, $f(BA) \to AC$ and so on, are unique.

The KTH (Schüldt et al., 2004) and DAVIS17 (Pont-Tuset et al., 2017) datasets are used to investigate the methods described above. KTH data provide a simple initial investigative platform with monochromatic videos featuring simple repetitious actions performed by humans (clapping, hand waving, punching, etc) against a static background while the DAVIS17 data features multi-channel RGB capture of generic action scenes across a variety of dynamic environments (surfing, breakdancing, paragliding, etc). See Figure 1 and Figure 2 for encoding examples using a simple autoencoder style network defined in (1) and applied according to eq. (2). Although compression and fidelity are peripheral considerations (see Wu et al. (2018) for full treatment), the examples presented generally yield faithful reproductions according to a multi-scale similarly measure (Wang et al., 2003) and achieve respectable compression ratios of 10x - 15x simply by modulating the number of convolutional layers in the CNN. Compression ratios can be further improved by applying mixed precision training techniques (Micikevicius et al., 2017).

A variety of practical engineering considerations arise during implementation. For example, it is possible to analyze sequence loss during post-production and inject a ground truth frame at a particular deterioration threshold to renew the recall process; analogous to the use of key-frames in standard compression algorithms. It is also possible to train additional networks which upscale the base reproduction. This upscaling approach has advantages in controlling network parameter growth (and hence the compression ratio) as two convolution layers each with 32 filters have only half the parameters compared to a single layer of 64 filters. In this way, eq. 2 is written as the composition of two functions

$$\begin{aligned} y_{i+1} &= f_\theta(y_i) \\ y_i^* &= f_{\theta_*}(y_i) \end{aligned} \tag{3}$$

where $f_\theta$ advances the sequence and the subsequent application of $f_{\theta_*}$ bolsters fidelity. From this point of view, $f_{\theta_*}$ performs the role of *denoising autoencoder* (Vincent et al., 2008) which strives to

---

[2]In some ways, dilation is akin to the order of an autoregressive model

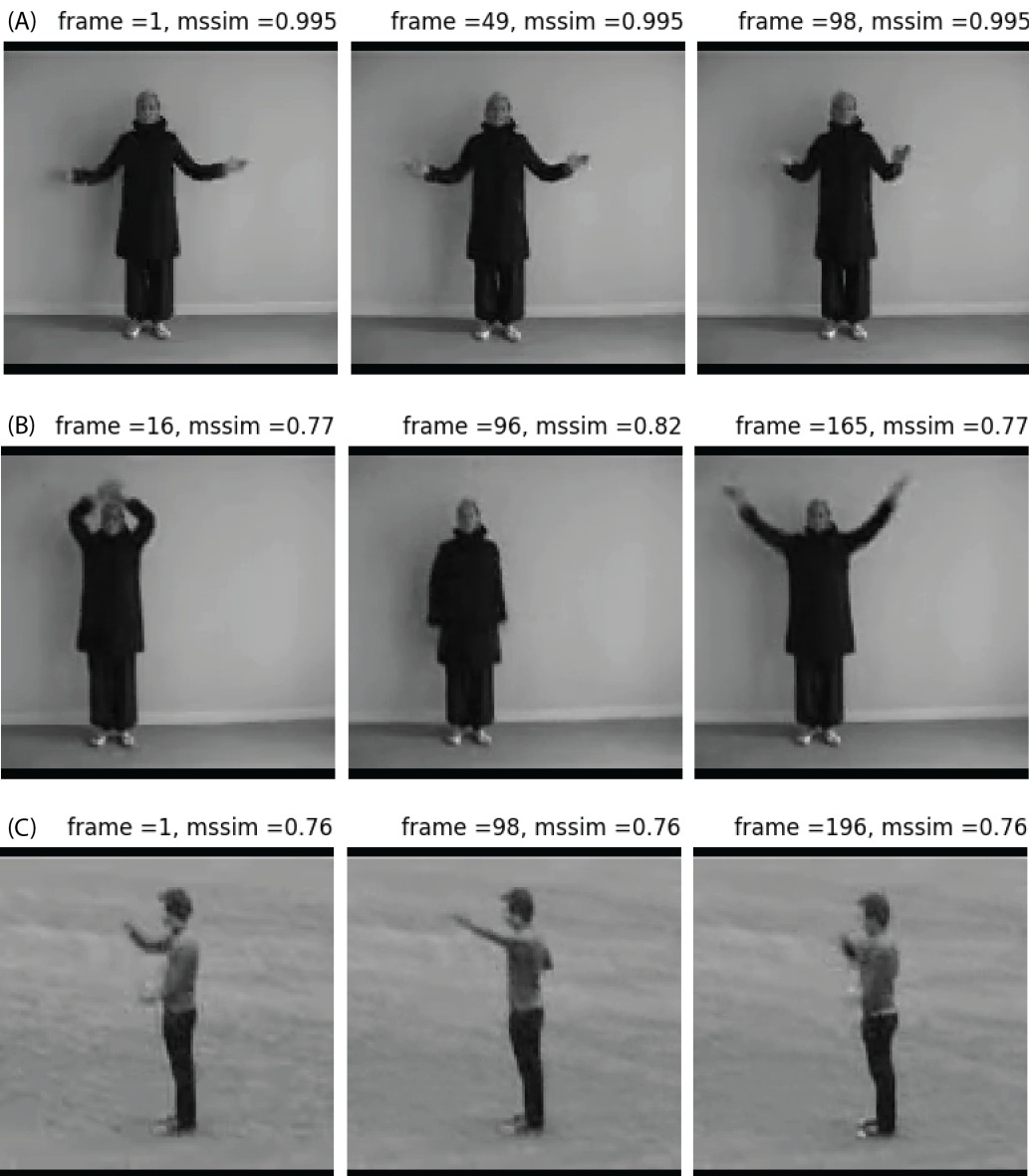

Figure 1: Video encoding reproductions at 128x128 pixel resolution from the KTH dataset. (A) A reproduction of *person15_handclapping_d4* using 128 convolutional filters per layer which, although not parameter efficient, achieve high fidelity after 99800 training epochs at full FP32 precision. (B) A reproduction of *person15_handwaving_d4* using only 32 convolutional filters per layer and dilation $d = 2$ after 363947 training epochs with mixed FP16 precision. This encoding achieves a compression ratio of 27x but at the cost of some fidelity. (C) Finally, a reproduction of *person25_boxing_d1* using 32 convolution filters per layer and dilation $d = 2$ after 473701 training epochs with FP16 precision.

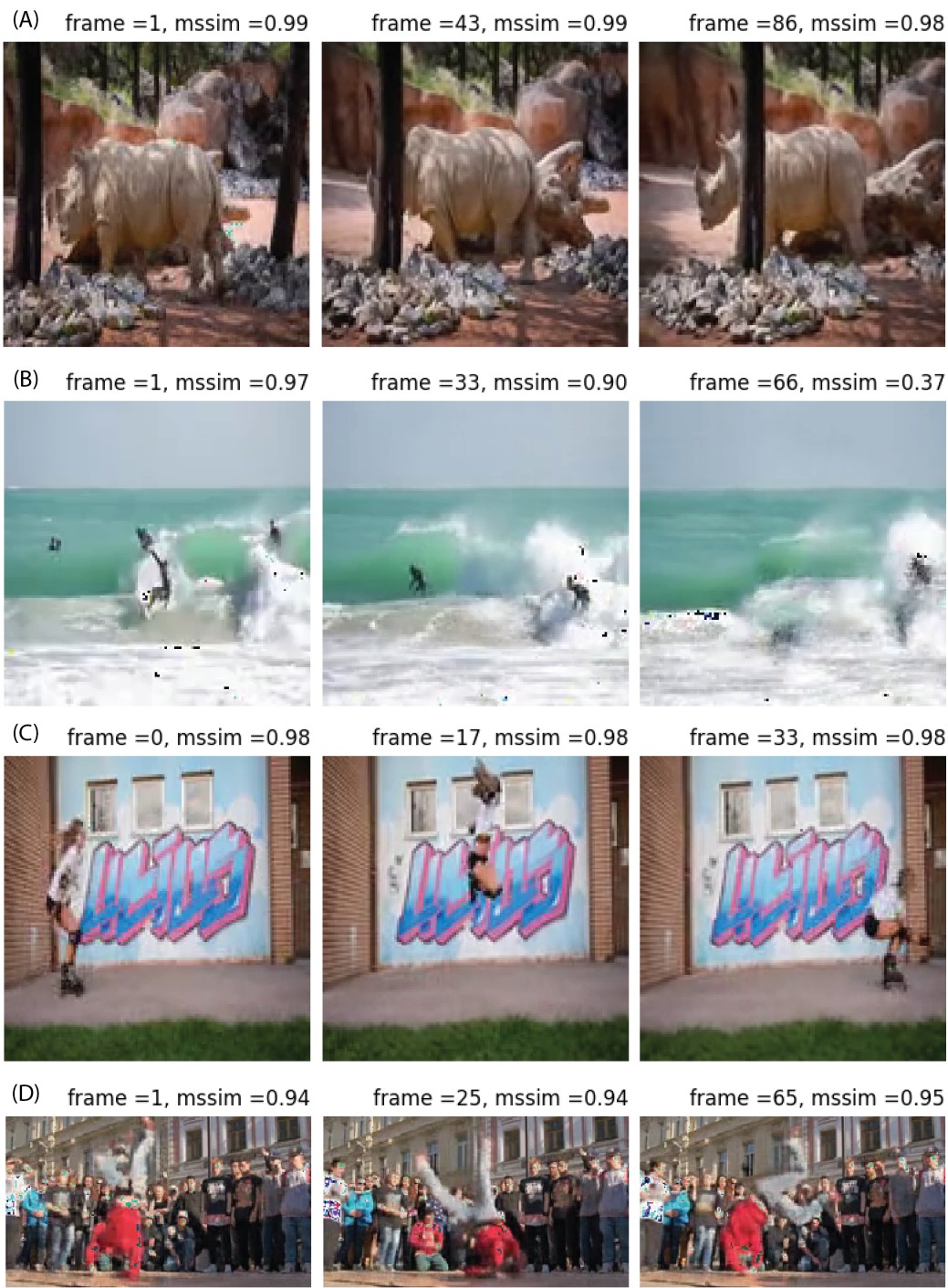

Figure 2: Video encoding reproductions from the DAVIS'17 dataset. (A) Rhino scene reproduction at 3x128x128 pixel resolution with 128 convolutional filters per layer and 200000 training epochs. (B) Miami surf scene recalled at 3x128x128 pixel resolution using 128 filters per layer and 77000 training epochs. (C) Recalled frames at 3x128x128 pixel resolution from the Rollerblade scene using 128 convolutional filters per layer and 250000 training steps. (D) Breakdance reproduction at 120p with pixel resolution 3x120x213 using 48 convolutional filters per layer at 85000 training epochs.

**Algorithm 1** Pseudo definition of the autoencoder style network used throughout this work and applied as defined in eq. (2). All layers define $nconv$ convolutional filter channels of size $3 \times 3$. The first and final layers takes $C \times d$ number of input and output channels respectively where $C$ is the number of image channels and $d$ is the sequence dilation. Where necessary, *stride* and *padding* are used to maintain dimensional equality of input and output tensors.

```
function NETWORK(nconv)
    x = relu(Conv2d(x))
    x = relu(Conv2d(x))
    x = relu(Conv2d(x))
                                                        ▷ upsample
    x = relu(Conv2d(x))
    x = relu(ConvTranspose2d(x))
                                                        ▷ upsample
    x = relu(Conv2d(x))
    x = ConvTranspose2d(x)
    return x
```

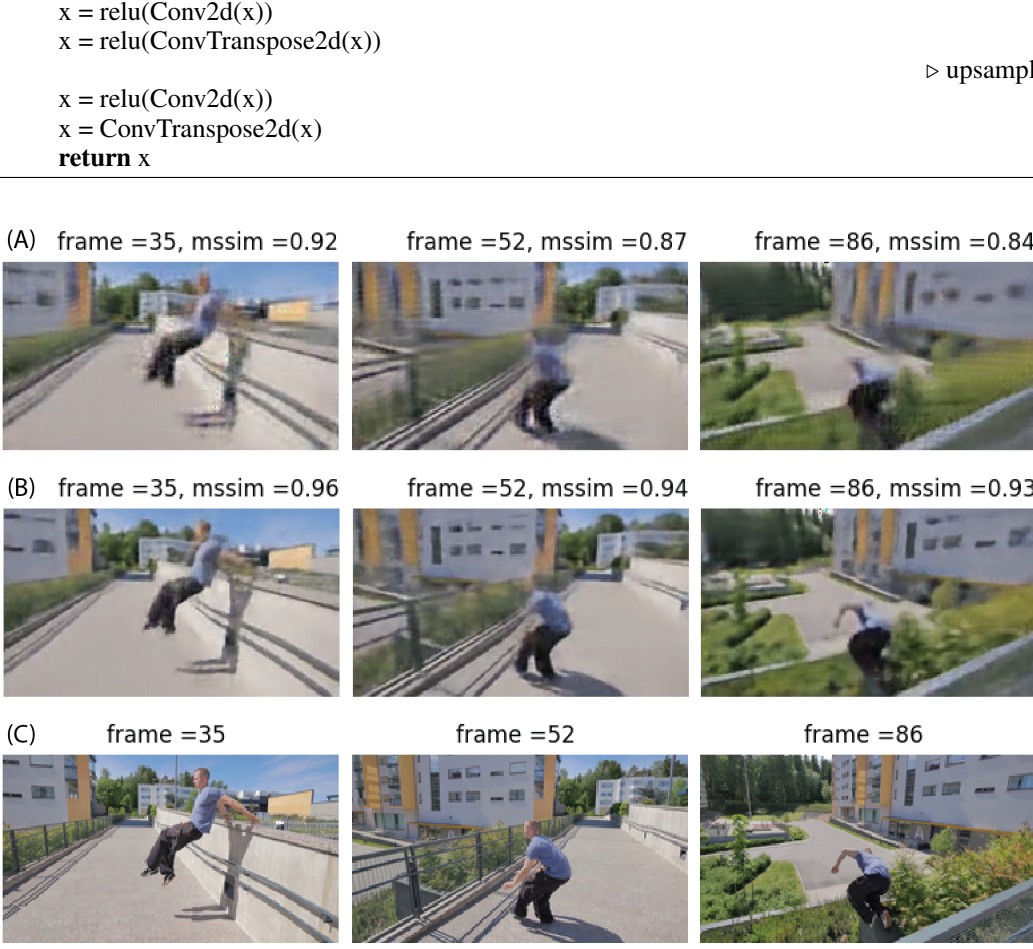

Figure 3: Video encoding reproduction of the DAVIS'17 Parkour scene at 120p using the upscaling approach described by eq. 3 with $M = 1$. (A) Frames, $y_i$, produced by the base network $f_\theta$, with 32 convolution filters per layer, which advances the sequence at each application. (B) Upscaled frames, $y_i^*$, produced by denosing autoencoder, $f_{\theta_*}$, with 32 convolution filters per layer. (C) Associated ground truth frames, $x_i$.

form a 'repaired' output from 'corrupted' input. This allows computational demands during recall to be modulated on a frame-by-frame basis according to platform limitations, real-time constraints, and so on. This application of upscaling is shown in Figure 3.

From here, we can generalize further and leverage the recent advancements in *normalizing flows* whereby a simple initial density is transformed into a more complex one by applying a sequence of transformations until a desired level of complexity is attained (Tabak & Turner, 2013; Dinh et al., 2014; Jimenez Rezende & Mohamed, 2015; Kingma et al., 2016). The density $q^K(z)$ obtained by

successively transforming a random variable $z^0$ with distribution $q^0$ through a chain of $K$ transformations $f_k$ is:

$$z^K = f_K \circ \ldots \circ f_2 \circ f_1(z^0) \qquad (4)$$

where eq. (4) is shorthand for the composition $f_M(f_{M-1}(\ldots f_1(z^0)))$. In this way, the initial density is said to 'flow' through the sequence of mappings. For our purposes, the original sequence $X$ defined in (1) can be seen as just an arbitrary random noise process in some high-dimensional space. Rather than model the process $X$ directly, instead, it can be approximated, to some acceptable level of fidelity, through a chain of transformations over some known latent process, $Z = \{z_0, \ldots, z_N\}$. That is, given a random process, $Z$, defined by $g$, we have

$$z_{i+1} = g(z_i)$$
$$z_i^j = f_j \circ \ldots \circ f_2 \circ f_1(z_i) \qquad (5)$$

where $z_i^j \approx x_i$ represents $j$ consecutive transformations applied to $z_i$, and where each $f_j$ is a deep neural network defined by Algorithm 1. These transformations are simply a series of bijective mappings (under the given restricted domain and range) and therefore alleviate the aforementioned sequence training challenges associated with realizing computations of the form defined by eq. (2). The question then becomes, what is $g$?

The choice of $g$ must be some random but deterministic high dimensional process with the requirement that $\forall z_i, z_j \in Z, z_i \neq z_j$. That is, the elements of $Z$ are unique such that the process is noncyclic in order to support a bijective mapping. A key insight is that given some $g$ defined by Algorithm 1 and applied according to eq. (5), the process $Z$ is completely defined by the starting point, $z_0$. Therefore, from the point of view of Kolmogorov complexity, $\{z_0, g\}$ is a random computation for the production of the sequence, $Z$, and normalizing flows with differential programming provide the framework to transform this computation: $X \xleftarrow{f} \{z_0^0, g\}$. From a compiler point of view, $z^j$ is simply an abstract intermediate representation (IR).

A simple recipe for realizing $g$ is to choose any two random points $z_0, z' \in \mathbb{R}^{C \times H \times W}$ and learn parameters, $\phi$, such that $N$ recursive applications of $g_\phi$ starting with $z_0$ yields $z'$. Although, in practice, after sufficient number of training iterations of $g_\phi$, it does not matter if $z_N$ computed according to eq. (5) does not equal $z'$ rather it is only required that $g_\phi$ satisfy the noncyclic random walk properties described above. This recipe for $g$ is described in Algorithm 2 and demonstrated in Figure 4. The full workflow of transforming random process, $Z$, generated by $g$ into the target sequence $X$ according to eq. (5) is shown in Figure 5.

---

**Algorithm 2** Simple recipe for realizing a high-dimensional autoregressive noise process.

$\quad$ **function** FORWARD($g$,$z$,$n$) $\hfill \triangleright$ defined according to eq. (5)
$\quad\quad$ **for** $i \leftarrow 0, n$ **do**
$\quad\quad\quad$ $z \leftarrow g(z)$
$\quad\quad$ **return** z

$\quad$ $N \leftarrow$ length($X$)
$\quad$ $z_0 =$ uniform($C \times H \times W$) $\hfill \triangleright$ samples from a uniform distribution over the interval $[0, 1)$
$\quad$ $z' =$ uniform($C \times H \times W$)
$\quad$ $g_\phi \leftarrow$ NETWORK($nconv$) $\hfill \triangleright$ described in Algorithm 1

$\hfill \triangleright$ determine $\phi$ that minimizes the $\|z - z'\|^2$

$\quad$ **for** $i \leftarrow 0, T$ **do**
$\quad\quad$ $z \leftarrow$ FORWARD($g_\phi, z_0, N$)
$\quad\quad$ $l \leftarrow \|z - z'\|^2$
$\quad\quad$ $\delta \leftarrow \frac{\partial l}{\partial \phi}$
$\quad\quad$ $\phi \leftarrow$ step($\phi$,$\delta$)

---

## CRYPTOGRAPHIC CONSIDERATIONS

The neural video encoding process shares many analogs with symmetric cryptographic algorithms. In general, *cryptography* is the body of techniques for secure communications focused on the con-

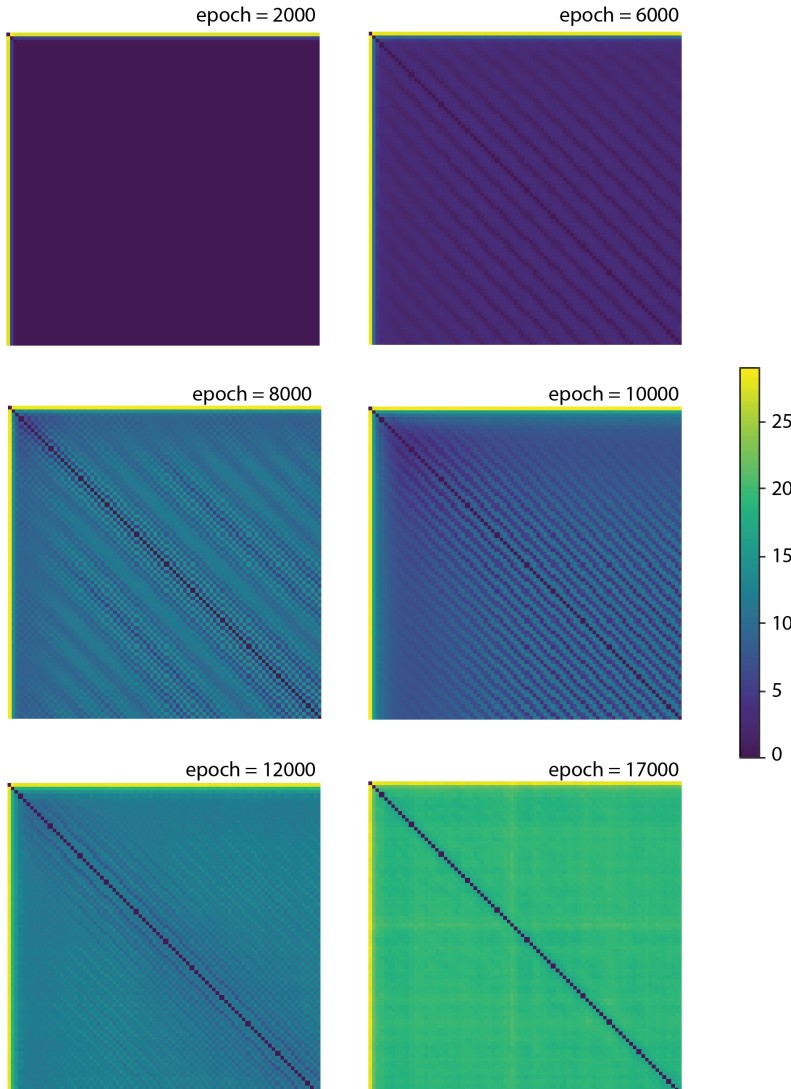

Figure 4: Visualizing the training of a high-dimensional autoregressive noise process. The computation, $g$, as defined is a simple convolutional neural network according to Algorithm 1 and trained as defined by Algorithm 2. In this instance, $g_\phi$ was initialized with $nconv = 8$. To help visualize the learning process, distance matrices are calculated over the elements of $Z$ at various stages of training. After 17000 epochs of training, the distance matrix confirms that the elements generated by $g_\phi$ are sufficient to support bijective transformation.

struction of protocols that prevent third parties or the public from reading private information. A *protocol* defines a system of information exchange which includes rules, syntax, semantics and synchronization of communication. Moreover, a cryptographic protocol usually focuses on various aspects of information security such as confidentiality, data integrity and authentication. Cryptography is often synonymous with *encryption* which is the conversion or encoding of information from a coherent readable state to an incoherent state of apparent randomness and the reverse process of *decryption*. A *cipher* is defined by a pair of algorithms specifying an encryption and decryption process controlled by the use of an auxiliary private key which is required to decrypt the associated data. A key must be selected before using a cipher to encrypt data and without the key, it should be extremely difficult or impossible to decrypt the contents. When the same key is used for both

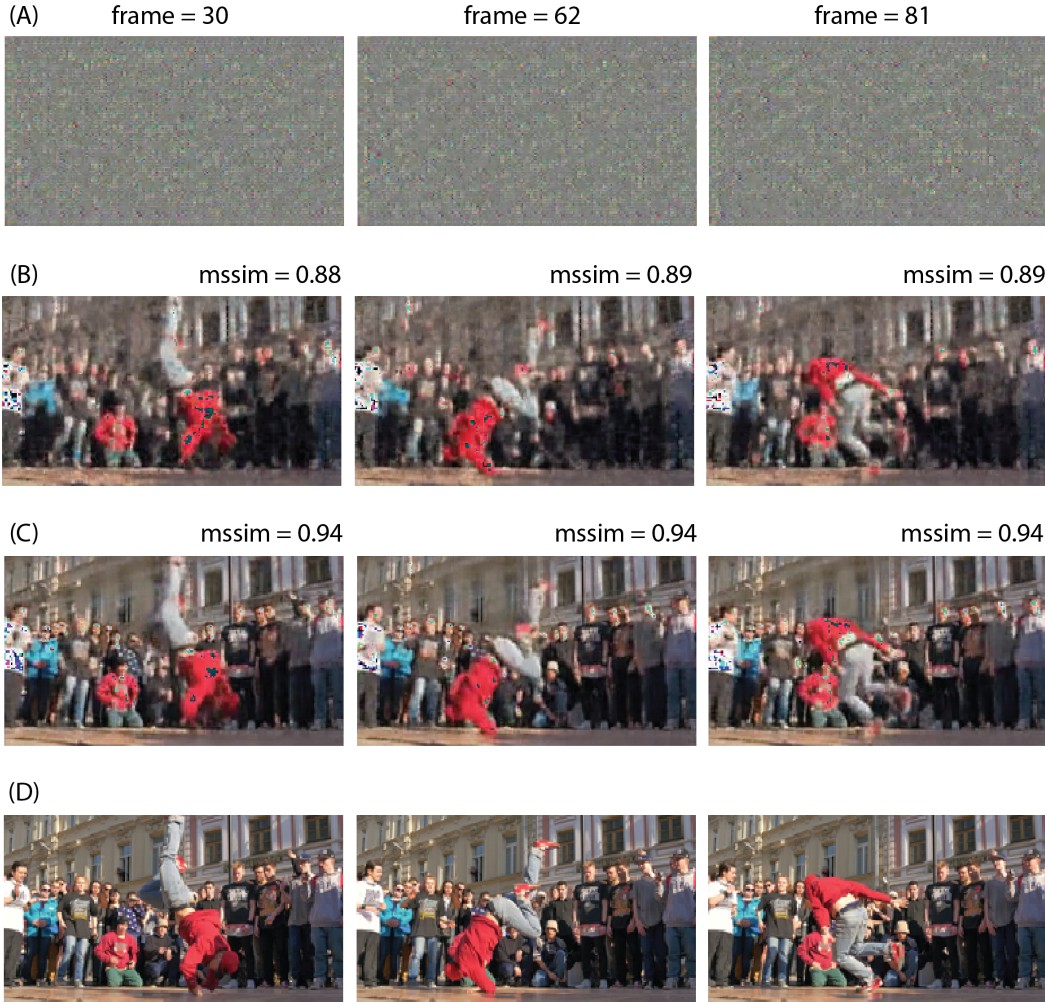

Figure 5: Neural video encoding via normalizing flows of the DAVIS'17 Breakdance scene at 120p. (A) Visualizations of the variables $\{z_{30}^0, z_{62}^0, z_{81}^0\}$ from some random autoregressive noise process, $g_\phi$, having $nconv = 8$ and procured according to Algorithm 2 having 10000 training epochs. (B) The variables $\{z_{30}^1, z_{62}^1, z_{81}^1\}$ as computed by eq.(5) where $z_i^1 = f_{\theta_1}(z_i^0) = f_{\theta_1} \circ g_\phi(z_{i-1}^0)$ and $f_{\theta_1}$ is configured with $nconv = 32$ having been trained for 30000 epochs. (C) The variables $\{z_{30}^2, z_{62}^2, z_{81}^2\}$ computed as $z_i^2 = f_{\theta_2}(z_i^1) = f_{\theta_2} \circ f_{\theta_1}(z_i^0)$ with $f_{\theta_2}$ again having $nconv = 32$ and trained for 30000 epochs. (D) Ground truth sequence elements $\{x_{30}, x_{62}, x_{81}\}$ from the unencoded data, $X$

encryption and decryption, the cipher defines a *symmetric key algorithm*.[3] Symmetric cryptography deals with the efficient construction of pseudo random functions which form the building blocks of symmetric algorithms and have the form $x = h(z)$ with the following properties (Ramkumar, 2014)

1. Given $x = h(z)$, small changes to $z$ produce output $x'$ unrelated to $x$.

2. Given $z$, there is no way to make reliable predictions regarding any part of $x$.

3. Given $x$, the easiest way to find $z'$ that satisfies $x = h(z')$ is brute-force search.

4. The fastest way to determine preimages $x$ and $x'$ satisfying $h(x) = h(x')$ is brute-force search.

---

[3]See Paar & Pelzl (2010), Schneier (2015), and Ferguson et al. (2010) for additional discussion.

From a high-level perspective, it is easy to identify latent variable $Z$ and target sequence $X$ as the encrypted and unencrypted data respectively, while the variable, $z_0^0$ performs the role of shared key. The encryption algorithm is defined by the realization of a generator function, $g$, which consumes $z_0^0$ to produce $Z$, and the decryption algorithm is defined by the application of learned transformations, $F = \{f_j\}_{j=1}^M$, applied according to eq. (5). Together, the neural video components, $\{g, F\}$, form a *neural cipher* and with shared use of key, $z_0^0$, define a symmetric cryptographic algorithm. An informal analysis indicates the necessary alignment with the properties of pseudo random functions. For a particular instance of $g_\phi$, changes to $z_0^0$ yield unrelated sequences $Z'$ and, by definition, the random values of $Z$ alone provide no information regarding $X$. Moreover, as black box functions, given some target element $x_i$, the only way to find $z'$ satisfying $x_i = F(z')$ is indeed brute-force search. It is natural to go further and consider the significantly more empowered situation in which an adversary could perform differential calculations over $F$ to optimize the search for $z'$ which minimizes $\|x_i - F(z')\|$. Effectively, this means the decryption algorithm and a portion of the un-encrypted data have been compromised. Even so, without the associated generator, $g$, it remains very difficult or impossible to resolve additional content since despite having obtained $z'$, it is not possible to make reliable predictions regarding any part of $x$. Finally, an example of these properties is provided in Figure 6 demonstrating a failed attempt to decrypt an encoded video with an imposter key. In this situation, both the encryption and decryption algorithms have been compromised but the adversary has no knowledge of $Z$ or $X$. A robust cryptographic algorithm must remain secure even if everything about the implementation, except the key, becomes available to an adversary.[4] Indeed, some work has suggested that CNNs retain accurate image information with different degrees of geometric and photometric invariance(Mahendran & Vedaldi, 2014). However, the authors attempts at extracting meaningful content using *total variation* based approaches were unsuccessful since having the encryption and decryption algorithms alone provides little basis for the construction of a loss function to minimize (see also Ulyanov et al. (2017)). This discussion is, of course, not intended to provide a thorough cryptographic analysis of neural video encoding but rather to facilitate understanding of various properties and highlight the potential for information security.

## RELATED WORK

The storage of information in neural networks has a rich history (Kohonen & Lehtiö, 1989). Early parallel models of associative memory were used in the investigation of long-term memory systems in cognitive science probing questions of how information is represented in the brain and what kinds of processes operate on it (Anderson & Hinton, 1989). Some of this early work was motivated by *holography* as a mechanism for distributed associative recall (Willshaw, 1989) made possible by Nobel prize wining advancements in the development of the holographic method by Gabor. Interestingly, as highlighted by Hinton (1989) in the same body of work, Marr, Palm, and Poggio (Marr et al., 1978) were particularly interested in equations of the form $C^{n+1} = \sigma(L(C^n))$ where $L$ is a linear operator and $\sigma$ a nonlinear function and stressed the importance of understanding their behavior. These associative memory models continue to garner some attention (Karbasi et al., 2013; Mazumdar & Rawat, 2015)

Next frame prediction in moving imagery and video data shares similar motivations with neural video encoding but differ in ultimate objective. See Lotter et al. (2017) for discussion and relevant references. Although observed but not highlighted, video encoding networks trained on KTH data for simple repetitious actions often exhibited the ability to generalize beyond encoded sequence for short periods and so demonstrating a limited capacity for what could be argued as generalization versus pure recall.

The application of neural networks to cryptographic problems is not new (Kanter et al., 2002; Klimov et al., 2002; Abadi & Andersen, 2016) and the synergies between machine learning and cryptography have been appreciated for some time (Rivest, 1993). Many applications of machine learning to cryptography are realized through cryptanalysis tools and majority of neural network applications have been orchestrated at the bit level and around anti-symmetric public protocols as in (Kanter et al., 2002). Neural video encoding defines a simple symmetric protocol operating *in situ* at the pixel and hence leverages the native advantages of distributed representation. Although,

---

[4]This is known as *Kerckhoffs's principle* and (Claude) *Shannon's maxim*

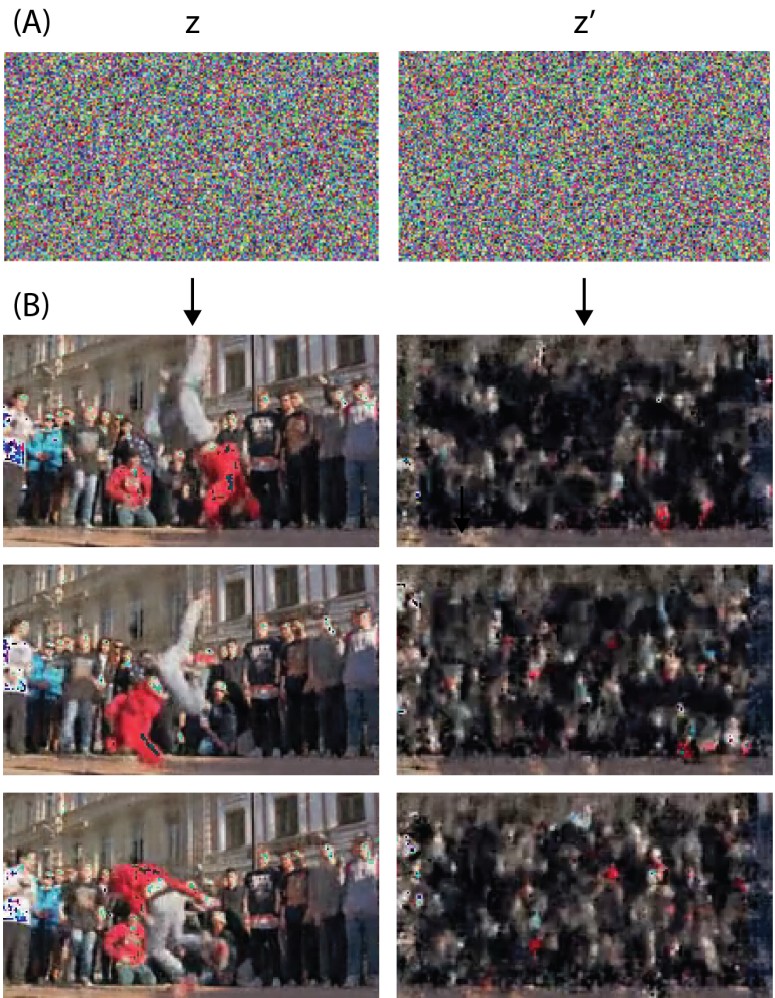

Figure 6: An attempt to decrypt (i.e. recall) an encoded video using an imposter key. (A) Key $z_0^0$ and imposter key $z_0'^0$. (B) On the left, $z_0^0$ and $g_\phi$ applied to generate the appropriate keys $\{z_{30}^0, z_{62}^0, z_{81}^0\}$ and decrypted through application of $F$. On the right, the imposter key, $z_0'^0$, with the same $g_\phi$ yields an invalid key stream $\{z_{30}'^0, z_{62}'^0, z_{81}'^0\}$ which decrypt to incoherent states under $F$.

visual cryptographic methods applied to images directly have existed for some time as well (Naor & Shamir, 1995) (see also Punithavathi & Subbiah (2017) for recent survey).

## CONCLUSION

This work demonstrates encoding methods for the storage and retrieval of video data with convolutional neural networks. There remain many open considerations but ultimately the ability to realize computations of arbitrary complexity which represent data is clear. Moreover, these computations are abstract representations of data and therefore exhibit desirable synergies with cryptographic systems underpinning confidentiality to facilitate information security. Recent advancements in all-optical neural networks (Lin et al., 2018; Zuo et al., 2019) motivate the future exploration of combining past holographic based associative memories efforts with the encoding methods presented here to create potential new forms of optical memory and beyond.

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
