# OpenReview forum: "Neural Video Encoding"
_ICLR.cc/2020/Conference — Reject_

### Official Review · AnonReviewer1 · 2019-10-15
**Official Blind Review #1**

**Rating:** 1

**Review:**

The authors show that CNNs are somewhat able to compress entire videos within their parameters that can be reconstructed by an autoregressive process. This is an interesting idea that has been explored in a different context before (e.g., Deep Image Prior, Ulyanov et al. 2017). There is also plenty of work in the area of exploiting NNs for video compression/encoding (see [1] for instance). However, a bit unusual is the choice of compressing a video into network parameters, which is quite an expensive process using backprop. I could not find any motivation for why this would be a good idea, potentially because the paper does not state any explicit goal/contribution. In any case the authors show merely some mediocre qualitative results. There are no comparisons to prior work and no empirical results. The combination of methods also seems a bit arbitrary.

Therefore this paper has no valuable contribution in its current form and I vote for rejection.

Major issues
- No empirical results
- No comparison to any baseline or prior work
- Confusing: the paper has almost no structure, e.g., there are almost no sections at all. Symbols are not used consitently (e.g., functions f and g).

Questions
- What's the goal of employing normalizing flows in this paper?

[1] Image and Video Compression with Neural Networks: A Review. Ma et al. 2019.


**Experience Assessment:**

I have published one or two papers in this area.

**Review Assessment: Checking Correctness Of Derivations And Theory:**

N/A

**Review Assessment: Checking Correctness Of Experiments:**

I assessed the sensibility of the experiments.

**Review Assessment: Thoroughness In Paper Reading:**

I read the paper at least twice and used my best judgement in assessing the paper.

---

### Official Review · AnonReviewer3 · 2019-10-21
**Official Blind Review #3**

**Rating:** 1

**Review:**

I'm not sure what the contribution of this paper is. It seems to contain a variety of weakly related motivational examples.  The paper begins by stating the computer vision advanced due to "ability to search for suitable task algorithms via differentiable programming." But I don't think this claim is reasonable, i.e., I don't think CNNs are searching for task algorithms.

Then the paper explains Kolmogorov complexity, but this is completely unrelated to the rest of the paper. At no point in this paper is this concept used and this motivation is not useful.

The paper then introduces the method, which is a standard, simple autoencoder. Some image evaluations of this are shown, but no contribution in terms of the model is made.

Finally, the paper briefly mentions some connections to cryptography, but it is unclear what or why this connection matters.

The paper has no real experimental analysis, it doesn't seem to propose anything new or make any clear contribution, and overall, is quite unclear on what it is trying to do. The current paper is unsuitable for publication.

**Experience Assessment:**

I have published in this field for several years.

**Review Assessment: Checking Correctness Of Derivations And Theory:**

N/A

**Review Assessment: Checking Correctness Of Experiments:**

N/A

**Review Assessment: Thoroughness In Paper Reading:**

I read the paper thoroughly.

---

### Official Review · AnonReviewer2 · 2019-10-26
**Official Blind Review #2**

**Rating:** 3

**Review:**

Summary:
This paper introduces convolutional neural networks to encode video sequences. In particular, it exploits an autoencoder style convolutional network to compress videos with certain ratios by modulating the number of convolution layers. Furthermore, it studies an autoregressive model to learn the sequential information over videos.

Strength:
The idea of using convolutional networks to encode the videos is interesting.
The used autoencoder and autoregressive techniques are promising for video encoding.


Weakness:
The paper gives me an impression that there are very few works to apply convolutional networks (including autoencoder with autoregressive processing) to encode videos. I cannot see any existing works sharing the same motivation, and the paper does not evaluate any related works for comparison. But it seems the use of autoencoder and autoregressive techniques look very straightforward to me. Please clarify the concern.

About technical contribution, it is not clear to see if there are any improvement over the conventional autoencoder and autoregressive models. It would be better to make some necessary discussions in the paper. In addition, it seems no descriptions showing how to learn the bijective mappings f_1, ..f_j.

Regarding the evaluation, the paper uses visual results and multi-scale SSIM (mssim) to study the effectiveness of the proposed method on the KTH and DAVIS’17 datasets. While a small number of video frames are used, it would be necessary to evaluate the overall (average) mssim on the whole dataset.

There are quite a few typos or grammatical problems on sentences such as “...algorithms which encode sequences of video image frames.”, “f is some convolutional neural network…”, “analogous to the use of key-frames in standard compression algorithms.


**Experience Assessment:**

I do not know much about this area.

**Review Assessment: Checking Correctness Of Derivations And Theory:**

I assessed the sensibility of the derivations and theory.

**Review Assessment: Checking Correctness Of Experiments:**

I carefully checked the experiments.

**Review Assessment: Thoroughness In Paper Reading:**

I read the paper at least twice and used my best judgement in assessing the paper.

---

### Decision · Program_Chairs · 2019-12-19

**Decision:**

Reject

**Comment:**

The paper has several clarity and novelty issues.